# In Vivo Investigation of the Effect of Dietary Acrylamide and Evaluation of Its Clinical Relevance in Colon Cancer

**DOI:** 10.3390/toxics11100856

**Published:** 2023-10-13

**Authors:** Christiana M. Neophytou, Andromachi Katsonouri, Maria-Ioanna Christodoulou, Panagiotis Papageorgis

**Affiliations:** 1Tumor Microenvironment, Metastasis and Experimental Therapeutics Group, Basic and Translational Cancer Research Center, Department of Life Sciences, European University Cyprus, 2404 Nicosia, Cyprus; c.neophytou@euc.ac.cy; 2State General Laboratory, Ministry of Health, 2081 Nicosia, Cyprus; akatsonouri@sgl.moh.gov.cy (A.K.); mar.christodoulou@euc.ac.cy (M.-I.C.); 3Tumor Immunology and Biomarkers Group, Basic and Translational Cancer Research Center, Department of Life Sciences, European University Cyprus, 2404 Nicosia, Cyprus

**Keywords:** acrylamide, colon cancer, acrylamide-induced carcinogenesis, food contaminants, dietary exposure, RNA metabolism, pathway analysis

## Abstract

Dietary exposure to acrylamide (AA) has been linked with carcinogenicity in the gastrointestinal (GI) tract. However, epidemiologic data on AA intake in relation to cancer risk are limited and contradictory, while the potential cancer-inducing molecular pathways following AA exposure remain elusive. In this study, we collected mechanistic information regarding the induction of carcinogenesis by dietary AA in the colon, using an established animal model. Male Balb/c mice received AA orally (0.1 mg/kg/day) daily for 4 weeks. RNA was extracted from colon tissue samples, followed by RNA sequencing. Comparative transcriptomic analysis between AA and mock-treated groups revealed a set of differentially expressed genes (DEGs) that were further processed using different databases through the STRING-DB portal, to reveal deregulated protein–protein interaction networks. We found that genes implicated in RNA metabolism, processing and formation of the ribosomal subunits and protein translation and metabolism are upregulated in AA-exposed colon tissue; these genes were also overexpressed in human colon adenocarcinoma samples and were negatively correlated with patient overall survival (OS), based on publicly available datasets. Further investigation of the potential role of these genes during the early stages of colon carcinogenesis may shed light into the underlying mechanisms induced by dietary AA exposure.

## 1. Introduction

Acrylamide (AA) has been linked with various toxicological effects such as neurotoxicity, reproductive toxicity, immunotoxicity and carcinogenicity [1,2,3]. AA was classified as “probably carcinogenic to humans (group 2A)” by the International Agency for Research on Cancer (IARC) [4]. The main route of exposure to AA is through diet. High AA levels are formed during cooking of many commonly consumed foods including French fries, potato chips, breakfast cereal and coffee [5,6]. Smoking and occupational exposure to AA have also been evaluated [7,8,9]. Because the AA molecule is small and hydrophilic, it can easily reach every organ and virtually every tissue in the body [10]. For this reason, theoretically all tissues are potential targets for carcinogenesis.

Evidence indicates that AA causes cancer in laboratory animals. The genotoxic mechanism of AA is well established. AA is metabolized to its epoxide metabolite glycidamide (GA) by cytochrome P4502E1 [11]. GA forms DNA adducts and is genotoxic [12]. However, possible non-genotoxic pathways of AA-induced carcinogenesis are still under investigation. In mice and rats, tissue-specific carcinogenicity after exposure to AA has been observed. Increased occurrence of mammary gland tumors was detected in rats which received AA through drinking water [13]. AA may exert a carcinogenic role on select body sites by affecting hormonal balances. A positive association between dietary AA intake and estrogen and progesterone receptor-positive breast cancer risk in women has been reported [14]. 

Epidemiologic data on AA intake in relation to cancer risk are limited and contradicting. Positive associations between dietary AA intake and the risks for development of endometrial, ovarian, estrogen receptor-positive breast, renal cell cancers and lung cancer have been observed [15,16]. When AA is consumed orally, the gastrointestinal tract is exposed to considerable amounts of this substance [11]. AA absorption is affected by other food components such as proteins that bind AA and decrease its final concentration in tissues [17]. An increased risk of esophageal cancer, based on 341 cases, emerged in subjects with intermediate levels (≥24 µg/day) as compared to low AA (<24 µg/day) intake [18]. However, one study reported no association between dietary AA and gastrointestinal cancer risk [19]. 

In this study, we investigated the gene expression profile changes in the colon of mice following oral administration of AA. Mice received AA through oral gavage for 4 weeks and colon tissue was then processed and analyzed by whole-genome sequencing. Using computational tools, we identified a set of genes that were upregulated (*Rps9*, *Rps14*, *Rps15*, *Rps17*, *Rps24*, *Rps27a*, *Rpl4*, *Rpl11*, *Rpl13a*, *Rpl14*, *Rpl18*, *Rpl24*, *Rpl36*, *Rpl39*, and *Eif4a2*) or downregulated (*Upf3a*, *Smg6*, *Ddx23*, *Ppie*, *Sptb*, *St3gal3*, *Gnai2*, *Lmo7*, *Capza1*, *Sec24a*, *Hnrnpd*, *Furin*, *Bcl10*, *Dcun1d5*, and *Spsb2*) in the AA-treated compared to the mock-treated group. Data derived from the REACTOME database suggest that the majority of these genes are implicated in protein and RNA metabolism. Based on freely accessible databases (UALCAN, TNMplot, Human Protein Atlas, and Kaplan Meyer plotter) we also validated which of these genes are highly expressed in primary colon cancer samples in humans compared to normal colon tissue and we evaluated their correlation with overall survival (OS) of colon cancer patients. Our results provide important insights to the molecular pathways affected by dietary AA in the gastrointestinal tract in vivo and may be used in the future to fully characterize the underlying mechanisms of this chemical to the colon, which may ultimately lead to carcinogenesis. 

## 2. Materials and Methods

### 2.1. In Vivo Experiment

Male Balb/c mice (6 weeks old, *n* = 5/group) were randomized to one mock-treated and one AA-treated group. The AA-treated mice (average weight 20 g each) received oral gavage of 0.1 mg/kg AA, diluted in 100 μL of PBS, daily for 4 weeks. The mock-treated group received PBS (phosphate-buffered saline) in equal quantities. The AA dose was selected based on previous studies in mice and rats, where 0.1 mg/kg of AA delivered by oral gavage was sufficient to expose tissues to a significant amount of AA and its metabolite GA [20,21]. At the end of the experiment, tissue samples from the gastrointestinal tract (GI) were obtained from mice following euthanization. The excised tissues did not show signs of visible abnormality. Enzymatic digestion solution was prepared in serum-free medium, DMEM, containing 0.2 mg/mL Collagenase, 2 mg/mL Dispase, 100 U/mL DNase I and was pre-heated to 37 °C. Using a scalpel blade, tissue was diced into pieces approximately 1–2 mm in size and added to 10 mL of enzymatic digestion solution in a 15 mL falcon tube. The falcon tube was then placed on a rotating platform inside a 37 °C incubator (Auxilab, Navarra, Spain) for 35 min. Fetal Calf Serum (FCS) was added to a final concentration of 10% and filtered through a 40 µm filter. Any undigested pieces were forced through the filter using the plunger from a syringe. Cells were pelleted at 500× *g* for 5 min and total RNA was extracted with Trizol reagent (Invitrogen, Carlsbad, CA, USA) following the manufacturer’s protocol. RNA was cleaned-up using the RNeasy Mini kit (Qiagen, Hilden, Germany) and the samples were sent to “ATLAS Biolabs GmbH” company (Berlin, Germany) for RNA sequencing, as a subcontracted service. All in vivo experiments were conducted in accordance with the animal welfare regulations and guidelines of the Republic of Cyprus and the European Union under a license acquired by the Cyprus Veterinary Services (CY/EXP/PR.L1/2021), the Cyprus national authority for monitoring animal research.

### 2.2. Bioinformatics Analysis

The FastQ files containing the RNA sequencing data were processed for gene expression analysis using the Galaxy platform (https://usegalaxy.org, accessed on 16 June 2023) [22]. The pipeline utilizing the Hisat, HTseq, and DeSeq2 tools was applied to produce normalized gene-expression counts in each sample. Differential expression between groups was assessed using Student’s *t*-test. For the selection of differentially expressed genes (DEGs), the following cut-offs were applied: false discovery rate (FDR) < 0.05 and fold-change > 2 (up-regulated genes) or <0.5 (down-regulated genes). The FDR is the ratio of the number of false positive results to the number of total positive test results. It was calculated using the formula (*p*-value of the *t*-test × rank position of the *p*-value among the total number of tests)/number of tests. Up- and down-regulated genes were further processed using Gene Ontology (GO) enrichment, REACTOME, Kyoto Encyclopedia of Genes and Genomes (KEGG) database analysis and other applications to reveal deregulated protein-interaction networks, molecular pathways, biological processes and functions, and hubs proteins.

### 2.3. Protein–Protein Network

Protein–protein interaction networks provide information on the molecular framework of cellular processes. Gene-set enrichment analysis (GSEA) was performed using the STRING tool (Search Tool for the Retrieval of Interacting Genes/Proteins) [23]. Regulated molecular pathways were retrieved from the REACTOME database [24]. Deregulated pathways were considered those retrieved with a false discovery rate (FDR) < 0.05. 

### 2.4. TNMplot

The TNMplot is an online platform that allows the comparison of gene expression levels between normal, tumor and metastatic samples using publicly available transcriptome-level datasets. The platform uses data generated by either gene arrays from the Gene Expression Omnibus of the National Center for Biotechnology Information (NCBI-GEO) or RNA-seq from The Cancer Genome Atlas (TCGA), Therapeutically Applicable Research to Generate Effective Treatments (TARGET), and The Genotype-Tissue Expression (GTEx) repositories. Statistical significance is computed using Mann–Whitney or Kruskal–Wallis tests. False Discovery Rate (FDR) is computed using the Benjamini–Hochberg method. The reliability of the database to provide differential gene expression in colon tissue samples was validated using equally sized training and test sets, at an FDR below 10%. The online analysis platform enables unrestricted mining of the database [25]. 

### 2.5. The Cancer Genome Atlas (TCGA) Data Validation

The University of ALabama at Birmingham CANcer (UALCAN) data analysis Portal (https://ualcan.path.uab.edu/, accessed on 4 July 2023), enables a comparison of gene expression in normal and primary colon adenocarcinoma samples [26,27]. UALCAN is a comprehensive online resource that provides easy access to publicly available cancer OMICS data (TCGA, MET500, CPTAC and CBTTC) and allows users to validate the clinical relevance of potential genes of interest. The website also provides graphs and plots depicting gene expression profiles in normal and cancer tissues. For our analysis, we focused on colon adenocarcinoma vs. normal colon tissue. We used the TCGA tool, specifically for colon adenocarcinoma and searched for the expression levels of individual genes. The graphs show the expression levels in transcripts per million in normal versus primary tumor samples. 

### 2.6. Human Protein Atlas Database for Protein Expression

The Protein Atlas project (Human Protein Atlas proteinatlas.org, accessed on 16 June 2023) is an open-access knowledge resource that shows the distribution of proteins across all major tissues and organs in the human body [28]. We derived representative images of immunohistochemistry-stained normal vs. colon and adenocarcinoma tissue. Intensity and expression estimates (low, medium, and high) as well as antibody numbers are provided in the database.

### 2.7. Kaplan–Meier Plotter Analysis

Kaplan–Meier plotter (https://kmplot.com, accessed on 16 June 2023) [29], an in silico online tool, was used to predict the overall survival of colon cancer patients based on a meta-analysis of expression levels of candidate genes. RNA sequencing gene expression data from multiple annotated colon cancer studies are combined into a single database from which we queried for associations between expression of selected genes and predicted patient outcomes. Sources for the databases include GEO, EGA, and TCGA. The graphs indicate the correlation between the expression levels of selected genes or the mean expression of multiple genes with overall patient survival (OS). The primary purpose of the tool is a meta-analysis-based validation of these genes as survival biomarkers in human samples [30]. KM Plotter was used to study the prognosis value for *RPS9*, *RPS14*, *RPS15*, *RPS17*, *RPS24*, *RPS27A*, *RPL4*, *RPL11*, *RPL13A*, *RPL14*, *RPL18*, *RPL24*, *RPL36*, *RPL39*, and *EIF4A2* in rectum adenocarcinoma samples (*n* = 165) relevant to overall patient survival (OS). The log-rank *p*-values were calculated and presented in the graphs. 

## 3. Results

### 3.1. Differential Gene Expression and Pathway Analysis in AA-Exposed Colon Tissue

Treated mice were administered AA orally for a period of 4 weeks, followed by colon tissue harvesting, total RNA extraction and sequencing. Notably, none of the excised murine colon tissues exhibited obvious signs of malignant lesions or any other visible abnormality. The identified differentially expressed genes between mock- and AA-treated groups were used to construct and visually reveal protein-interaction networks and hubs proteins (Figure 1). Analysis revealed that proteins involved in RNA processing are significantly upregulated in AA-treated tissue. These proteins are implicated in Nonsense Mediated Decay (NMD) enhanced by the Exon Junction Complex (EJC) responsible for mRNA degradation, gene mapping to L13a-mediated translational silencing of Ceruloplasmin expression, the SRP-dependent co-translational protein targeting to membrane pathway as well as the major pathway of rRNA processing in the nucleolus and cytosol. In addition, proteins implicated in the formation of the ribosomal subunits and in translation were found to be differentially expressed in AA-treated colon tissue. Genes *Rps9*, *Rps14*, *Rps15*, *Rps17*, *Rps24*, *Rps27a*, *Rpl4*, *Rpl11*, *Rpl13a*, *Rpl14*, *Rpl18*, *Rpl24*, *Rpl36*, *Rpl39*, and *Eif4a*, which were significantly upregulated following treatment (fold change > 2, *p* < 0.05), were found to be implicated in most activated pathways (Figure 1B). Moreover, genes *Upf3a*, *Smg6*, *Ddx23*, *Ppie*, *Sptb*, *St3gal3*, *Gnai2*, *Lmo7*, *Capza1*, *Sec24a*, *Hnrnpd*, *Furin*, *Bcl10*, *Dcun1d5*, and *Spsb2*, also implicated in the above pathways, were found to be significantly under-expressed (fold change < 0.5, *p* value < 0.05) as compared to the untreated control group.

Based on the above analysis, we also composed a table showing the significantly differentiated genes that were common among most identified pathways affected by the treatment (Table 1).

### 3.2. Evaluation of Clinical Relevance of Affected Genes Using Publicly Available Human Datasets

Following the pathway analysis described above and based on the current published literature, we selected genes that may be relevant to AA-induced carcinogenesis. Specifically, using human databases, we investigated the following genes: *RPS9*, *RPS14*, *RPS15*, *RPS17*, *RPS24*, *RPS27A*, *RPL4*, *RPL11*, *RPL13A*, *RPL14*, *RPL18*, *RPL24*, *RPL36*, *RPL39*, and *EIF4A2* using the TNMplot, UALCAN, Human Protein ATLAS and KMplotter online tools for meta-analysis.

To investigate whether the identified genes may serve as possible biomarkers of AA-induced colorectal tumorigenesis, we investigated their mRNA expression levels in normal and cancer colon tissue. The TNMplot database allows for differential gene expression analysis in tumor, normal, and metastatic tissues. The multiple gene analysis tool provides an overview of the selected gene set in colon adenocarcinoma tissue using RNA-Seq-based data (Figure 2).

In addition, using the UALCAN data analysis portal, we compared the mRNA expression levels of selected genes in normal colon and primary colon adenocarcinoma (COAD) samples. We used the TCGA tool, specifically for colon adenocarcinoma and searched for the expression levels of individual genes. The graphs show the expression levels in transcripts per million in primary COAD tumor samples (*n* = 286) versus normal tissue (*n* = 41). Selected genes include *RPL11, RPS15*, *RPS27A*, *RPL18* (statistical significance < 1 × 10^−12^), *RPS17* (statistical significance < 1.11022302462516 × 10^−16^) and *RPL36* (<1.62458935193399 × 10^−12^) (Figure 3). The rest of the significantly differentially expressed genes are shown in Appendix A. 

The expression pattern of these genes was then further validated by comparing protein expression levels in normal versus cancer colon tissue. Representative images showing immunohistochemical detection of the selected proteins were derived from the Human Protein Atlas website (Figure 4). RPS15, RPL11 and RPL36 displayed medium expression in normal endothelial colon cells, whereas RPL18 had low expression in normal colon. On the other hand, RPL11 and RPL18 proteins had moderate intensity in colon tissue, whereas RPS15 and RPL36 showed strong staining intensity in colon cancer tissue. 

The list of differentially expressed genes implicated in protein synthesis was filtered using an online tool which performs a meta-analysis of publicly available microarray datasets and RNA sequencing data from patients with colon cancer to generate Kaplan–Meier survival curves. During this analysis, patients were separated into two groups on the basis of the expression levels of each gene, and the probability of OS over time was calculated. We found that higher expression of *RPS15*, *RPL11*, *RPL18* and *RPL36* could individually predict worse overall survival (OS) of patients with colon cancer compared to the low expression of these genes (Figure 5). 

Finally, based on this evidence, we hypothesized that since these genes are implicated in similar pathways related to ribosomal biosynthesis and protein translation, *RPS15*, *RPL11*, *RPL18* and *RPL36* expression levels may coordinately predict the survival of patients with colon cancer. Indeed, Kaplan–Meier plotting analysis revealed that patients with higher combined expression of *RPS15*, *RPL11*, *RPL18* and *RPL36* are more strongly associated with shorter OS compared to the expression levels of these genes individually (Figure 6).

## 4. Discussion

Epidemiological evidence on AA exposure and the development of different cancer types is limited and contradicting. This highlights the importance of unraveling the carcinogenic pathway initiated by AA exposure. 

Since diet seems to be the primary source of exposure to AA, many studies have focused on the effects of AA in the colon [31,32,33,34]. 

In our study, we used a dose of 0.1 mg/kg AA solution administered daily by oral gavage. This was the same as previous studies in mice and rats where the toxicokinetic analysis revealed that, at this concentration, AA was rapidly absorbed from oral dosing, widely distributed to tissues and was efficiently converted to GA [20,21]. This dose, albeit much higher than the estimated daily AA intake from typical human diets (0.5–4 μg/kg), is much closer to human exposure than those previously reported [35,36,37] In addition, epidemiological studies investigating dietary AA exposure do not take into account the accumulative exposure from smoking and other sources; therefore, the average intake may be underestimated for smokers and those individuals exposed to second-hand smoke. In addition, children are exposed to twice as much AA intake compared with the total population [38,39]. The AA dose plays a role in potential cancer-inducing molecular pathways. In vitro exposure of human primary hepatocytes to high levels of AA induced oncogenes with growth-promoting potential compared to lower concentrations that activated genes involved in the elimination of the toxicant [40]. However, in vivo studies in rats showed that higher doses may lead to metabolism saturation and decrease the percentage of GA formed [41]. 

The genotoxic mechanism of AA is well established. AA is metabolized to its epoxide metabolite glycidamide (GA) by cytochrome P4502E1 [11]. Both compounds form DNA adducts and are genotoxic. However, possible non-genotoxic pathways of AA-induced carcinogenesis are still under investigation. We identified a set of genes that may be implicated in the early stages of AA-induced carcinogenesis in the colon including *Rps9*, *Rps14*, *Rps15*, *Rps17*, *Rps24*, *Rps27a*, *Rpl4*, *Rpl11*, *Rpl13a*, *Rpl14*, *Rpl18*, *Rpl24*, *Rpl36*, *Rpl39*, and *Eif4a2.* Importantly, the pathway analysis revealed that the majority of DEGs (11 out of 14) are implicated in biological processes directly or indirectly involved in protein synthesis (Figure 1, Appendix A). The most important pathways identified are discussed in more detail below. 

Genes implicated in the Nonsense Mediated Decay (NMD) enhanced by the Exon Junction Complex (EJC) were found to be upregulated in AA-treated colon tissue (Figure 1A). Nonsense-mediated decay (NMD) is a gene expression regulation mechanism that degrades aberrant mRNAs carrying premature termination codons (PTCs) but also normal transcripts. It has been implicated in the pathophysiology of many human genetic diseases; in cancer, it has both pro- and anti-tumorigenic roles [42]. Genes implicated in the NMD pathway, such as *Upf1*, *Upf2*, *Smg1*, *Smg6*, and *Smg7*, are expressed in higher levels in colorectal cancer (CRCs) with microsatellite stability, and promote tumor growth in xenograft models [43]. 

Genes mapping to L13a-mediated translational silencing of Ceruloplasmin expression, another pathway that emerged from our study, have recently been implicated in prostate carcinogenesis. Specifically, biopsy and subsequent RNA sequencing of prostate cancer patients revealed that elevated expression levels of this genomic signature appeared in samples with higher Gleason scores [44]. 

Another pathway found to be significantly affected by AA exposure in our study is the SRP-dependent co-translational protein targeting to membrane. The signal recognition particle (SRP) is a ribonucleoprotein complex that plays a central role for protein delivery to the membrane or to secretory pathways. Interestingly, non-canonical functions of proteins involved in these processes have been revealed, such as a role in cellular stress response and modulation in apoptosis in autoimmune diseases. These newly discovered properties highlight a potential role of the deregulation of these pathways and the development of disease including cancer progression [45]. 

Many of the proteins found to be upregulated in AA-treated colon tissue are implicated in ribosome biogenesis and function. Ribosomes, consisting of a small 40S subunit (40S) and a large (60S) subunit, catalyze protein synthesis. The ribosomal subunits contain not only RNA but also approximately 80 structurally distinct proteins. The diagnostic and prognostic value of ribosomal proteins (RPs) in cancer has been recently highlighted [46,47]. It has been previously reported that ribosomal protein (RP) expression in colorectal carcinomas (CRC) is different from colorectal adenoma or normal mucosa [48]. The expression patterns of RPs correlate with the differentiation, progression or metastatic status of CRC. Individual RPs have also been associated with specific tumor phenotypes. Some of the proteins found to be upregulated in our study have known extra-ribosomal functions, including RPS9 (DNA repair), RPL4 (self-translation regulation), RPL11 (tumor suppressor gene regulation), and RPL18 (reviewed in [48]). 

We focused on four upregulated genes that we found to be clinically relevant based on various databases: *RPS15*, *RPL11*, *RPL18* and *RPL36.* Our analysis showcased *RPS15* as one of the most significantly overexpressed genes in colon cancer tissue and which correlates with worsened patient OS. This is supported by previous studies that implicate RPS15 in gastric cancer progression, proliferation and migration by affecting the Akt/IKK-β/NF-κB signaling pathway [49,50]. Recently, RPL11 was found to have a pro-tumorigenic role in NSCLC; RPL11 was highly expressed in NSCLC cells and promoted proliferation, migration, and cell cycle transition through the G1 phase [51]. A previous study reported that RPL18 is overexpressed in CRC tissue by interacting with double-stranded RNA (dsRNA)-activated protein kinase (PKR) and inhibiting dsRNA binding to PKR. The inhibition of PKR may enhance protein synthesis and cell growth in cancer [52]. Recently, it has been reported that a mutation in the RPL18 protein acts as a neo-epitope, eliciting a strong reaction from endogenous CD8 T cell responses in colorectal cancer mouse models; this could identify the mutant form of RPL18 as a potential target of immunotherapeutic strategies in CRC [53]. RPL36 that also displayed high levels in colon cancer (Figure 3 and Figure 4), has been previously reported to be involved in the early stage of hepatocarcinogenesis; it was expressed in 45 of 60 (75%) HCC by immunohistochemistry, but was not detected in corresponding non-tumors [54]. 

Our analysis revealed a specific transcriptional signature, implicated in the pathways described above, that are upregulated in colon tissue, that are differentially expressed following exposure to AA. Some of these genes have been previously reported to be involved in carcinogenesis. eIF4E, a protein regulated by mTOR, is the core component of the translation initiation complex that assembles at the 5′ cap of eukaryotic mRNAs. In non-proliferating cells, eIF4E is inhibited by the eIF4E-binding proteins (4EBPs); following stimulation of mTOR, 4EBPs are phosphorylated and blocked, leading to the activation of eIF4E [55]. MAPK is another upstream regulator of eIF4E [56]. The role of eIF4E as a key player in oncogenic transformation is supported by previous studies; phosphorylation of eIF4E occurs in breast cancer cells following exposure to growth factors and chemotherapy while its de-activation inhibited proliferation of lung and prostate cancer cells [57,58,59]. Activation of eIF4E was indispensable for the transformation of mouse models of cancer [60,61]. Importantly, its phosphorylated levels positively correlate with poor prognosis and disease severity in prostate and lung cancer in humans [62,63]. Since the majority of studies support the involvement of p-eIF4E in cancer progression, future studies should investigate the effects of AA on the phosphorylation status of this protein. Recently, high levels of eIF4A2 were associated with poor prognosis in esophageal squamous cell carcinoma [64]. 

RPL39, another protein found overexpressed in our study of AA-treated tissue, has been previously reported to be overexpressed in breast cancer. Knockdown of RPL39 in triple-negative breast cancer (TNBC) xenografts significantly inhibited primary tumor growth and metastasis [65]. RPL39 was also one of the most significantly upregulated (<1 × 10^−12^ in our TCGA analysis of adenocarcinoma vs. normal tissue samples (Figure 3). 

The RPS24 that we found to be upregulated in AA-treated colon tissue and significantly highly expressed in COAD tissue based on TCGA analysis (Appendix A) has been previously reported to promote colorectal cancer cell migration and proliferation in vitro [66]. Knockdown of RPS24 inhibited cell proliferation, colony formation, cell migration and induced S-phase cell cycle arrest. We also found that RPS27A was significantly elevated in AA-treated tissues and human cancer (Figure 1, Appendix A). Recently, it was reported that a high expression of RPS27A predicts poor prognosis in HPV type-16 cervical cancer patients [67]. RPS27A has also been reported to promote proliferation, regulate cell cycle progression, inhibit apoptosis and enhance chemoresistance in leukemia cells [68]. Finally, RPL24, which we found elevated across our analyses, is known to enhance translation and promote tumorigenesis; its functional inactivation through mutation suppressed colorectal cancer by promoting eEF2 phosphorylation via eEF2K [69]. Partial loss of RPL24 function is known to protect mice against Akt or Myc-driven cancers [70]. 

Contradicting our results, previous studies have shown that low expression levels of ribosomal proteins RPS9, RPS14, RPS27, RPL11 and RPL14 are related to a poor overall survival in breast cancer patients, especially in TNBC [71]. Therefore, the stage of cancer progression should be taken into account when assessing the differential expression of ribosomal proteins in tumorigenesis. *RPL13A*, that we found to be overexpressed in AA-treated mouse tissue, is considered a housekeeping gene with high transcriptional stability [72,73]. 

Notably, downregulated genes in AA-treated colon tissue include *Sptb.* SPTB was determined to be the single most discriminatory protein of NSCLC adenocarcinoma, displaying a 70% reduction in tumor tissue relative to control tissue, implying the dysregulation of membrane integrity [74]. LIM-domain only protein 7 (LMO7), also significantly downregulated in AA-exposed colon, has been suggested to act as a tumor suppressor for murine lung adenocarcinoma [75]. *CAPZA1* was downregulated by miR-875-5p in esophageal squamous cell carcinoma, causing a tumor-promoting function [76]. It has been previously reported that AA alters the miRNA profiles in multiple tissues of rats and that it induces human hepatocarcinoma cell proliferation through the upregulation of miR-21 expression [77,78]. Some of the downregulated genes include *Sec24a*, an essential mediator of ER-induced cell death following cell damage and ER stress [79]. It is plausible that the deregulation of genes implicated in DNA damage response and apoptosis, including *Bcl10* [80], may contribute to the initiation of cancer. 

## 5. Conclusions

In conclusion, our study revealed a different transcriptional signature in mouse colon tissue, following oral gavage administration of AA. We then correlated our data with findings in human studies regarding colon cancer. However, extrapolating our results to humans has some limitations: 1. Human cancer studies involving AA provide unreliable data when it comes to risk quantitation, so a direct comparison between animal and human data is challenging [35]. 2. Cancer development in rodents following AA exposure is affected by species-specific factors, as studies have reported differences in the metabolism of AA and lower internal exposure of its metabolite GA in humans [81]. 3. The dose used in our study is higher than the estimated dietary human exposure. However, there is mounting concern that chronic AA exposure may have cumulative effects and exposure to higher doses of AA for shorter durations may result in comparable toxicity [82,83,84].

In summary, PPI network construction (Figure 1) provided insights on how these genes interact and may be involved in the early stages of malignant transformation. Further analysis revealed that some of these genes, notably *rps15*, *rpl11*, *rpl18*, and *rpl36*, are significantly increased in primary adenocarcinoma tissue samples and co-ordinately predict for worse overall survival of colon cancer patients (Figure 3, Figure 5 and Figure 6). Finally, IHC-processed tissue images from the Human Protein Atlas database confirm higher protein expression levels encoded by these genes in colon cancer compared to normal tissue (Figure 4). Some of these genes have been previously shown to be implicated in carcinogenesis, but their exact mechanism of action is not fully elucidated. Our findings support a role of these genes in the development of pre-malignant lesions during the early stages of carcinogenesis; their combined expression may potentially act synergistically to function in a pro-tumorigenic fashion. We speculate that upregulation of genes implicated in protein translation may occur in pre-cancerous lesions, thus accelerating protein synthesis and cell proliferation. Future elucidation of their mechanistic role in the early stages of malignancy may further support their significance as valuable prognostic markers or therapeutic targets for colon cancer.

## Figures and Tables

**Figure 1 toxics-11-00856-f001:**
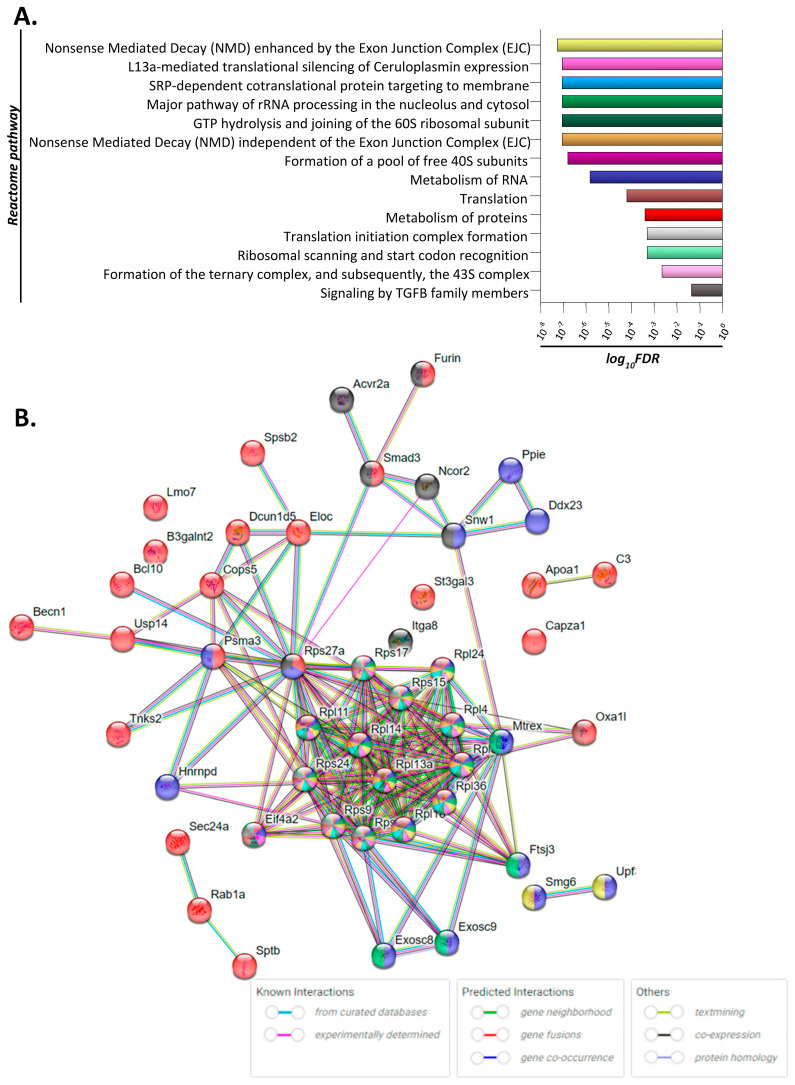
Deregulated molecular signaling in colon tissue of acrylamide-treated mice. (**A**) Bar diagram depicting the significance (log_10_FDR; false discovery rate) of the enriched REACTOME pathways/biological processes. Each pathway is labeled by different color. (**B**) Network of interactions involved in the significantly enriched pathways/biological processes. Different genes/proteins are involved in different (one or more) AA−affected pathways, as this is designated by the differently colored nodes. Edges (connections between nodes) represent protein−protein associations; either known interactions, predicted interactions or other associations as indicated by the key of interactions.

**Figure 2 toxics-11-00856-f002:**
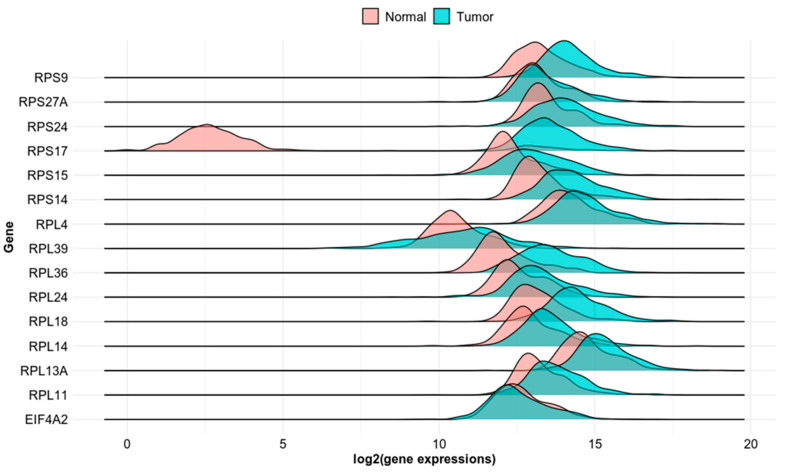
Differential expression of *RPS9*, *RPS27A*, *RPS24*, *RPS17*, *RPS15*, *RPS14*, *RPL39*, *RPL36*, *RPL24*, *RPL18*, *RPL14*, *RPL13A*, *RPL11* and *EIFA2* in normal and colon cancer tissue based on RNA-seq data.

**Figure 3 toxics-11-00856-f003:**
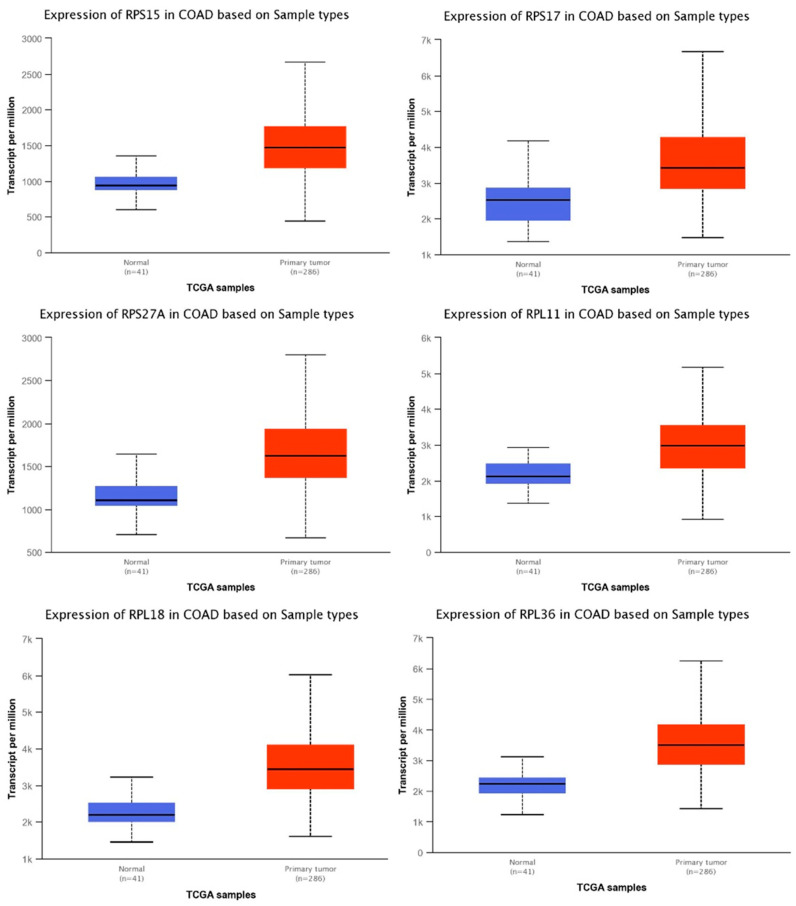
Levels of mRNA expression in human colon adenocarcinoma tissue (COAD) compared to normal cells. Selected genes include *RPS15*, *RPS17*, *RPS27A*, *RPL11*, *RPL18*, and *RPL36*. Figures generated based on UALCAN TCGA tool analysis, with criteria fold-change and *p*-values. Blue bar indicate the levels of the transcripts in normal tissues, while red bars indicate the levels of transcripts in primary tumor tissues.

**Figure 4 toxics-11-00856-f004:**
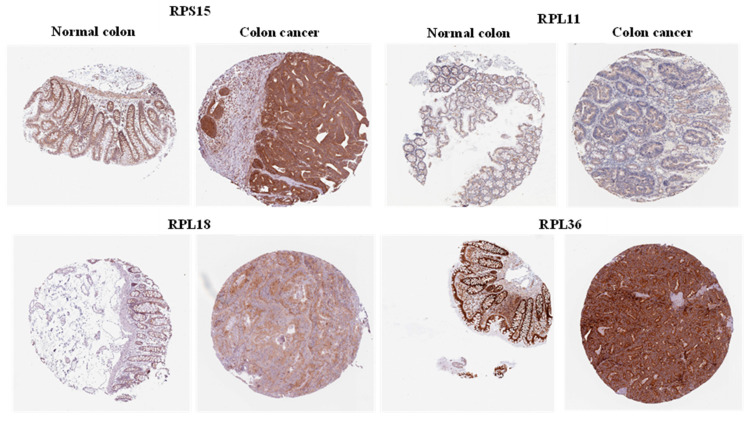
Protein expression of selected genes in normal colon and colon cancer tissues. A representative image for the protein expression levels of each gene was derived from the Human Protein Atlas database.

**Figure 5 toxics-11-00856-f005:**
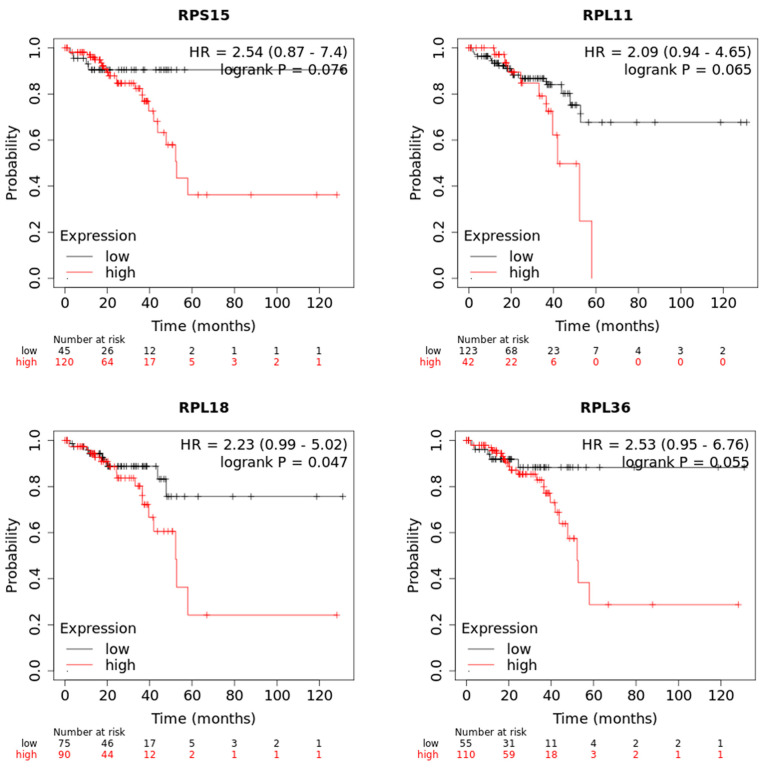
The mRNA expression levels of *RPS15*, *RPL11*, *RPL18* and *RPL36* are associated with poor survival of colon cancer patients. Kaplan−Meier survival analysis for assessment of overall survival (OS) based on tumor *RPS15*, *RPL11*, *RPL18* and *RPL36* expression in patients with colon cancer. Survival curves were generated by using the Kaplan−Meier Plotter online tool. Curves were compared by log-rank test.

**Figure 6 toxics-11-00856-f006:**
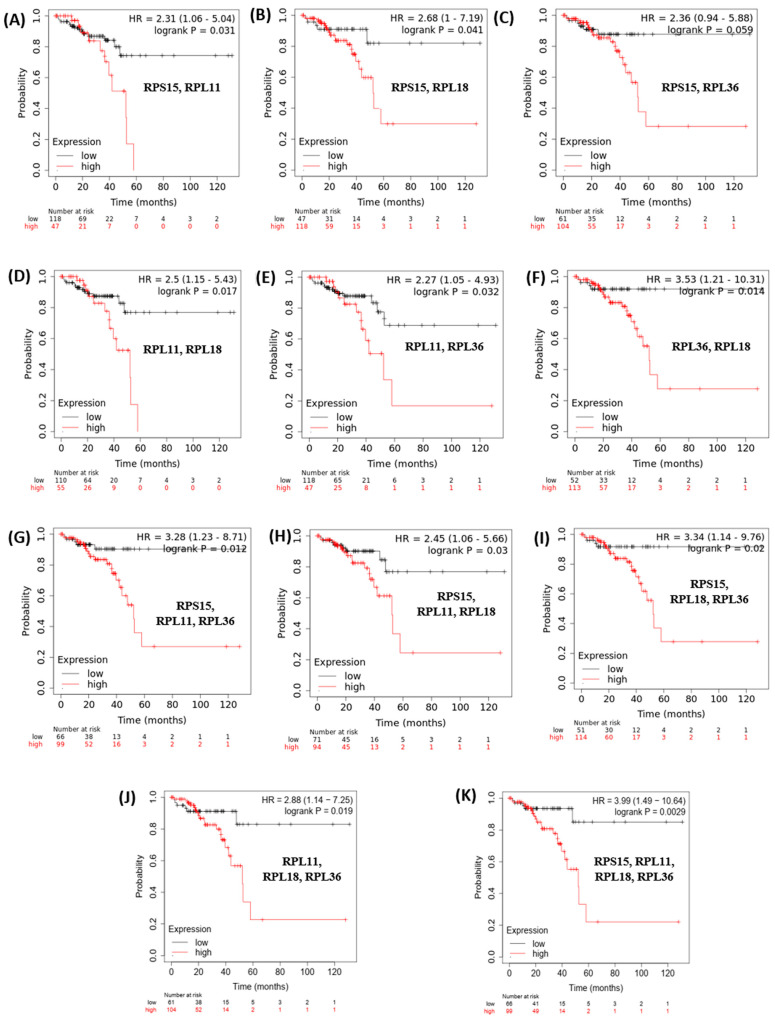
Increased combined mean mRNA expression levels of RPS15, RPL11, RPL18 and RPL36 are significantly associated with poor survival of colon cancer patients. Kaplan−Meier analysis for assessment of OS in patients with colon cancer, based on combined expression of RPS15, RPL11, RPL18 and RPL36. (**A**) RPS15, RPL11; (**B**) RPS15, RPL18; (**C**) RPS15, RPL36; (**D**) RPL11, RPL18; (**E**) RPL11, RPL36; (**F**) RPL36, RPL18; (**G**) RPS15, RPL11, RPL36; (**H**) RPS15, RPL11, RPL18; (**I**) RPS15, RPL18, RPL36; (**J**) RPL11, RPL18, RPL36; (**K**) RPS15, RPL11, RPL18, RPL36. Survival curves were generated by using the Kaplan−Meier Plotter online tool based on data stratified based on the median. Curves were compared by log−rank test.

**Table 1 toxics-11-00856-t001:** Fold change of gene expression following RNA sequencing analysis comparing mock- and acrylamide-treated (0.1 mg/kg) mice (*n* = 5/group) in colon tissue. Fold change was considered significant when ≥2 for up-regulated genes or ≤0.5 for down-regulated genes relative to the control/untreated group (*p* < 0.05).

Gene	Functional Role ^1^	Fold Change
Upregulated		
*Rps9*	Component of the 40S subunit	2.6
*Rps14*		2.4
*Rps15*		2.0
*Rps17*		3.0
*Rps24*		2.3
*Rps27a*		2.8
*Rpl4*	Component of the 60S subunit	2.3
*Rpl11*		3.7
*Rpl13a*		2.9
*Rpl14*		2.5
*Rpl18*		2.7
*Rpl24*		2.2
*Rpl36*		2.2
*Rpl39*		2.1
*Eif4a2*	- Peptide chain elongation - Activation of the mRNA upon binding of the cap-binding complex and eIFs, and subsequent binding to 43S	2.5
Downregulated		
*Upf3a*	- Peptide chain elongation - Structural constituent of nuclear pore	0.4
*Smg6*	Component of the telomerase ribonucleoprotein complex	0.48
*Ddx23*	Translation initiation, nuclear and mitochondrial splicing, and ribosome and spliceosome assembly	0.38
*Ppie*	Accelerates the folding of proteins.	0.46
*Sptb*	- Transport to the Golgi and subsequent modification - RAF/MAP kinase cascade	0.43
*St3gal3*	- Translation of structural proteins - Synthesis of substrates in N-glycan biosynthesis.	0.47
*Gnai2*	Hormonal regulation of adenylate cyclase	0.48
*Lmo7*	Signaling by ALK in cancer	0.41
*Capza1*	- Transport to the Golgi and subsequent modification - Golgi-to-ER retrograde transport	0.49
*Sec24a*	Mediates protein transport from the endoplasmic reticulum	0.49
*Hnrnpd*	Influences pre-mRNA processing and other aspects of mRNA metabolism and transport	0.40
*Furin*	Processes protein and peptide precursors trafficking through regulated or constitutive branches of the secretory pathway	0.42
*Bcl10*	Induces apoptosis and to activate NF-kappaB	0.47
*Dcun1d5*	- Cellular response to DNA damage stimulus - Positive regulation of protein neddylation - Regulation of cell growth	0.31
*Spsb2*	- Class I MHC-mediated antigen processing and presentation - Metabolism of proteins	0.45

^1^ Source: GeneCards human gene database.

## Data Availability

The data discussed in this publication have been deposited in NCBI’s Gene Expression Omnibus [85] and are accessible through GEO Series accession number GSE242206 https://www.ncbi.nlm.nih.gov/geo/query/acc.cgi?acc=GSE242206 (accessed on 10 October 2023).

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
