# Peer review of "In Vivo Investigation of the Effect of Dietary Acrylamide and Evaluation of Its Clinical Relevance in Colon Cancer"

_toxics, 2023, doi:10.3390/toxics11100856_

Round 1
Reviewer 1 Report
This manuscript presents the results of an interesting transcriptional analysis of acrylamide ingestion on cells of mouse colon. The presentation could be strengthened in several ways.
[1] Exposure was a daily dose of 0.1 mg/kg for 4 weeks. The choices of dose and time should be explained, since these could affect the results and their extrapolation to humans.
[2] The major results are provided in Table 1 as a list of gene expression changes compared to untreated control mice. Expression of differentially expressed genes (DEGs) was based on t-testing. Although false discovery rate (FDR) estimation is mentioned, how it was calculated is not given and should be. In addition, the authors do not mention whether statistically significant DEGs with a fold change <2 could be important.
[3] The finding that ribosomal proteins are upregulated is interesting. Figure 1 gives a protein interactome plot showing predicted relations between DEGs. This is difficult to interpret in part because the labels on the proteins are too small to read, and because the colors are difficult to match to the key. The bars connecting the spheres evidently are supposed to be specific colors, which is not obvious, and some of the types of interaction bars such as known and predicted interactions have the same color. How the graph relates to panel B should be clarified. The key has bars, but evidently the spheres have the colors of interest. The information presented is of uncertain value to the reader.
[4] The text states in section 3.1 (line 162) that none of the excised tissues showed signs of malignancy. Did they show any visible abnormality?
[5] The discussion compiles information from numerous studies that are not directly related to colon cancer, presumably because acrylamide targets more than this one tissue. While of some interest, the upshot is that the literature summarized has lots of studies with conflicting conclusions. The reader comes away without a clear picture. Thus the Discussion would benefit greatly from condensation by at least half and from providing a clearer analysis.
Minor points
[6] In the Intro (lines 43-44) and Discussion (303-304), the authors say that AA is metabolized to glycidamide and that both compounds form DNA adducts. This needs clarification. What are both compounds? Does AA form adducts without epoxidation?
[7] The two sentences in lines 47-49 are not connected. Is the intended meaning that mammary gland tumors are caused by hormonal imbalances induced by AA?
[8] In line 68, the text states that genes highly induced in primary colon cancer are studied. Are these DEGs or just highly expressed in tumors and in normal colon tissue?
[9] In section 2.4, the words “in real time” (line 117) are confusing. Does this mean the data become available as they are generated by the experimenters in their laboratories?
[10] In line 294, was the exposure to AA airborne?
[11] In line 306, SNP should be defined. If this refers to single nucleotide polymorphisms, the usage appears confusing there and in line 315. How does AA interact with a SNP?
[12] The fact that a mutation in a protein may act as a neo-epitope (line 377) does not clearly identify the gene as a target for immunotherapy (line 379), only specific mutations.
[13] This work concerns a transcriptional signature, not a gene signature (line 384).
Reviewer 2 Report
In the present manuscript authors have investigated the effect of dietary acrylamide on the clinical effects in colon cancer. I consider this information very useful and necessary to be able to establish a possible relationship between exposure to this contaminant in the diet and the development of colon cancer. However, I think the in vivo experiment should be described in more detail. It is important to mention how the dose of acrylamide administered to experimental animals has been estimated and whether that dose could be equivalent, lower or higher than dietary exposure in humans.
- Include justification of why that administered dose has been selected.
- How it was administered to the animals, in solution or solid, with food or not, etc.
- What was the weight of the animals (to calculate exposure per kg of body weight).
- Include in the discussion and conclusions a comment on these aspects, to estimate the possible extrapolation between the results with experimental animals and the real doses in the diet of the human population.
Reviewer 3 Report
The presented manuscript is extremely interesting both from a toxicological point of view and from a human safety or food production point of view. The manuscript is well prepared and worthy of publication. After reading the manuscript, I only have a few minor comments that could help the authors increase the scientific value of this text. I assume that the manuscript will be published in this journal after successful revision.
· Section „4. Discussion” - To what extent is it possible for the properties of AA to interact (stimulate or inhibit) with other food components? Does AA dose play a role in potential cancer-inducing molecular pathways?
· Section „5. Conclusions” – Please discuss the limitations of the research and observations received.
Minor editing of English language required.
Round 2
Reviewer 1 Report
The authors have carefully responded to each point of the critique. No further recommendations.
Author Response
We thank the Reviewer for the constructive criticism.